# The Prognostic Value of Derivatives-Reactive Oxygen Metabolites (d-ROMs) for Cardiovascular Disease Events and Mortality: A Review

**DOI:** 10.3390/antiox11081541

**Published:** 2022-08-09

**Authors:** Filippo Pigazzani, Davide Gorni, Kenneth A. Dyar, Matteo Pedrelli, Gwen Kennedy, Gabriele Costantino, Agostino Bruno, Isla Mackenzie, Thomas M. MacDonald, Uwe J. F. Tietge, Jacob George

**Affiliations:** 1MEMO Research, Division of Molecular and Clinical Medicine, University of Dundee, Dundee DD2 1GZ, UK; 2Division of Molecular and Clinical Medicine, Ninewells Hospital and Medical School, University of Dundee, Dundee DD2 1GZ, UK; 3Research and Development Department, H&D S.r.l., 43124 Parma, Italy; 4German Center for Diabetes Research (DZD), 40225 Neuherberg, Germany; 5Metabolic Physiology, Institute for Diabetes and Cancer (IDC), Helmholtz Diabetes Center, Helmholtz Zentrum München, 85764 Neuherberg, Germany; 6CardioMetabol Unit, Department of Laboratory Medicine and Department of Medicine, Karolinska Institutet, 17177 Huddinge, Sweden; 7Medicine Unit Endocrinology, Theme Inflammation and Ageing, Karolinska University Hospital, 17177 Stockholm, Sweden; 8Division of Population Health and Genomics, Ninewells Hospital and Medical School, University of Dundee, Dundee DD2 1GZ, UK; 9Department of Food and Drugs, University of Parma, 43121 Parma, Italy; 10Research and Development Department, Cor.Con. International S.r.l., 43124 Parma, Italy; 11Division of Clinical Chemistry, Department of Laboratory Medicine, Karolinska Institutet, 17177 Stockholm, Sweden; 12Clinical Chemistry, Karolinska University Laboratory, Karolinska University Hospital, 17177 Stockholm, Sweden

**Keywords:** oxidative stress, blood biomarkers, d-ROMs, prognostic value, cardiovascular diseases, mortality

## Abstract

Oxidative stress participates in the development and exacerbation of cardiovascular diseases (CVD). The ability to promptly quantify an imbalance in an individual reductive-oxidative (RedOx) state could improve cardiovascular risk assessment and management. Derivatives-reactive oxygen metabolites (d-ROMs) are an emerging biomarker of oxidative stress quantifiable in minutes through standard biochemical analysers or by a bedside point-of-care test. The current review evaluates available data on the prognostic value of d-ROMs for CVD events and mortality in individuals with known and unknown CVD. Outcome studies involving small and large cohorts were analysed and hazard ratio, risk ratio, odds ratio, and mean differences were used as measures of effect. High d-ROM plasma levels were found to be an independent predictor of CVD events and mortality. Risk begins increasing at d-ROM levels higher than 340 UCARR and rises considerably above 400 UCARR. Conversely, low d-ROM plasma levels are a good negative predictor for CVD events in patients with coronary artery disease and heart failure. Moreover, combining d-ROMs with other relevant biomarkers routinely used in clinical practice might support a more precise cardiovascular risk assessment. We conclude that d-ROMs represent an emerging oxidative-stress-related biomarker with the potential for better risk stratification both in primary and secondary cardiovascular prevention.

## 1. Introduction

Biological lifeforms have faced several major challenges during evolution on Earth, but the “*oxygen revolution*” represents one of the most dramatic [1]. When oxygen began to accumulate in the atmosphere, the shift to a strongly oxidising environment led to a mass extinction of ill-equipped lifeforms. The surviving species developed various mechanisms for sensing and counteracting the deleterious effects of reactive oxygen species (ROS), the highly reactive by-products of oxygen metabolism. Several systems have evolved in prokaryotic and eukaryotic cells to regulate ROSs, and today almost all organisms control free-radical levels by maintaining a fine equilibrium, coined reductive-oxidative (RedOx) balance [2,3,4].

In physiological conditions, ROSs are largely produced by metabolic reactions, and their effects are neutralised by an ensemble of enzymes and molecules endowed with antioxidant capacity. However, when ROS production exceeds the neutralising ability of antioxidant defences, a state of oxidative stress occurs, causing cell damage and death, and eventually organ dysfunction [2].

In the late 1950s, Harman proposed the “free radical theory of ageing” linking oxidative stress (OS) and an organism’s metabolic rate to life expectancy [5]. As oxidative stress has been linked to several human diseases, including cardiovascular diseases (CVD), Harman’s original hypothesis has evolved and been expanded [6,7,8]. Indeed, based on investigations over the last two decades (Figure 1) the role of oxidative stress in the pathophysiology of CVD is now well-accepted [8,9,10]. Therefore, with the advent of precision medicine [11], the measurement of oxidative stress and RedOx status might increase the existing network of clinical parameters currently used to characterise a patient’s state of health [2,8,12,13,14].

The derivatives-reactive oxygen metabolites (d-ROMs) test, which is based on the quantification of plasma organic peroxides, is an emerging biomarker for oxidative stress [8,15,16,17,18,19,20]. Unlike other oxidative-stress biomarkers [21], d-ROMs seem more suitable for use in clinical practice, being already available as a bedside point-of-care in vitro diagnostic (IVD), but also for the analysis of large population samples due to the relative simplicity of the method employed [15,16,17,18,19,20].

The aim of this review is to evaluate the most relevant published data available regarding the usefulness of blood d-ROMs levels as a potential prognostic biomarker for CVD events and mortality in patients with known and unknown CVD.

## 2. Literature Search Methods

A literature search was performed on PubMed and the Google Scholar engine (Figure 2), including articles published between 1 January 2000 and 31 December 2021. The following keywords were used: d-ROMs, cardiovascular diseases, myocardial infarction, heart failure, atrial fibrillation, cardiovascular death, cardiovascular mortality, all-cause mortality, and follow-up. In the case of acronyms, unabbreviated text and alternative naming without special characters (e.g., dROM instead of d-ROMs) were also used. Only articles published in the English language were included. References of the relevant published papers were also searched to help identify additional articles. Studies were excluded when subjects were not involved in follow-up to monitor the occurrence of cardiac events and mortality. After removal of duplicates and exclusion of studies where subjects were treated with antioxidant supplementation, we identified 12 relevant articles.

## 3. Reactive Oxygen Species (ROS), Antioxidants, and RedOx Balance

ROS are all biomolecules with unpaired electrons produced during cellular metabolism whose oxidising capacity depends on the oxygen redox potential. The most prominent ROS are the superoxide anion (O_2_^•−^), hydrogen peroxide (H_2_O_2_), and all peroxides (ROOR) [2,22].

The O_2_^•−^ is formed by the univalent reduction of triplet-state molecular oxygen (^3^O_2_). This reaction is promoted by enzymes such as NAD(P)H oxidase (NOX) and xanthine oxidase (XO), or by nonenzymatic reactive molecules such as ubiquinone (Coenzyme Q) in the mitochondrial electron transport chain. Under physiological conditions, O_2_^•−^ is enzymatically converted to H_2_O_2_ by the superoxide dismutase enzyme (SOD). However, when produced in large amounts, O_2_^•−^ can be converted by nonenzymatic pathways into H_2_O_2_, different types of organic peroxides (e.g., lipid peroxides), and singlet oxygen (^1^O_2_).

H_2_O_2_ has a cellular half-life long enough (1 ms) to allow its diffusion and propagation of oxidative processes to sites other than where it was generated [23]. Indeed, in the presence of positively charged transition metals (mainly Fe^2+^, Cu^+^), H_2_O_2_ (and other peroxides) can be converted into the highly reactive hydroxyl radical (^•^OH), alkoxy radical (^•^OR), and peroxidic radical (^•^OOR) by the Fenton reaction [24], further promoting oxidative damage [2,22].

ROS are constitutively present in cells and tissues in small but measurable concentrations. They act as guidance cues during embryonic development [25] and as signalling molecules activating relevant biological and physiological pathways [2]. However, an increase in ROS production, as during an altered metabolic state, or as a consequence of chronic exposure to external/environmental factors, may have harmful effects (Figure 3 and Figure 4, Appendix A) [13,26,27,28,29,30]. High free-radical levels enhance the oxidising capacity of a cellular or a tissue microenvironment, leading to oxidation of bio-macromolecules such as nucleic acids, lipids, and proteins. Oxidation alters the chemical structures of these compounds, leading to impairment of their physiological functions, which in severe cases culminates in tissue damages and diseases [2,6,7,8,12,22].

Lifeforms have developed the ability to counteract the deleterious effects of an excess of free radicals to varying degrees through antioxidant enzymes and molecules [2,4,7,22]. As defined by Halliwell and Gutteridge [31], antioxidants compete with oxidisable substrates, thus delaying or inhibiting their oxidation. Antioxidants are classified as direct/indirect and/or enzymatic/nonenzymatic depending on the mechanism through which they act (Appendix A). Hence, antioxidants are crucial for preserving the fine equilibrium between ROS production and degradation, known as RedOx balance, which represents a “key element” for life and maintenance of good health [2,22].

## 4. Role of Oxidative Stress in the Pathogenesis of Cardiovascular Diseases

Increased ROS levels induce inflammation, cell proliferation and migration, apoptosis, fibrosis, and extracellular matrix remodelling, which sequentially lead to alteration of vascular structures and functions [32,33]. A close correlation between oxidative stress and endothelial disfunction has been observed in systemic hypertension [34,35,36,37,38], hyperlipidaemia [39], and tobacco smoking [40].

The major players in ROS-mediated alteration are NAD(P)H oxidase (NOX), nitric oxide (NO) bioavailability, peroxynitrite ions (ONOO^−^), endothelial nitric oxide synthetase (eNOS), and oxidised LDL (Ox-LDL).

NO is the most abundant endothelial relaxing factor with potent antiatherosclerotic properties [41]; therefore, as a rule of thumb, conditions that result in a reduction in NO levels lead to adverse outcomes [34,35,36,42,43,44]. NO half-life depends upon ROS levels, especially on O^2•−^ levels, since it quickly reacts with NO to generate ONOO^−^, which is cytotoxic at high concentrations and causes oxidative damages to proteins, lipids, and nucleic acids. Moreover, it has been demonstrated that eNOS function is highly impaired by elevated levels of ONOO^−^ [34,37]. eNOS is an endothelial enzyme that produces NO and l-citrulline using O_2_ and l-arginine as substrates. It is a homo-dimeric enzyme, where each monomer is characterised by the presence of a reductase domain, containing NAD(P)H as cofactor and an oxidase domain, which uses haem and tetrahydrobiopterin (BH_4_) as cofactors. eNOS catalyses the electron transfer from NAD(P)H to the haem iron and to the BH_4_ co-factors in order to reduce and incorporate O_2_ into l-arginine and to produce NO and l-citrulline [37,45,46].

In ROS-mediated endothelial dysfunction, the limited availability of substrates and/or cofactors leads to the uncoupling of eNOS and in such situations the enzyme produces superoxide instead of NO, further generating ONOO^−^ [37,46]. Increased ROS levels also lead to the oxidation of BH_4_, an eNOS cofactor, further resulting in eNOS uncoupling. These mechanisms autopropagate, since ROS generated by eNOS uncoupling can further increase the level of oxidised BH_4_, thus exacerbating endothelial dysfunction.

In this context, where uncoupled eNOS contributes to increased ROS levels, LDL is oxidised to Ox-LDL, accelerating the development of atherosclerosis [38,42,47,48] by binding to its target receptor LOX-1 (Lectin-like oxidised low-density lipoprotein receptor-1). The Ox-LDL/LOX-1 complex is then internalised into endothelial cells triggering signalling pathways that promote (i) macrophage recruitment, activation, and transformation; (ii) activation of NOX; (iii) upregulation of the angiotensin-converting enzyme (ACE); and (iv) apoptosis through the activation of the mitogen-activated protein kinase (MAPK) signalling pathway [38,42,47,48]. Ox-LDL are internalised by macrophages via scavenger receptors (SRs), a process that is not feedback-inhibited by cellular cholesterol accumulation, and leads to foam cell formation and eventually cell death. Moreover, macrophages further contribute to endothelial dysfunction because of the release of proinflammatory cytokines, which recruit circulating cells of the immune systems. At a cellular level, the internalised Ox-LDL/LOX-1 complex also activates NOX and inhibits the synthesis of NO, contributing to the accumulation of O^2•−^ and other ROS. Moreover, the Ox-LDL/LOX-1 complex results in the upregulation of ACE, which induces the accumulation of angiotensin II, cell injury, alterations, and upregulation of LOX-1. It has been hypothesised that LOX-1 activation plays a crucial role in the connection between the renin–angiotensin system and dyslipidaemia usually observed in hypertensive subjects [38,42,47,48,49,50].

Hence, experimental evidence clearly supports that ROS-mediated oxidative stress induces deleterious effects by triggering a cascade of signals that results in endothelial dysfunction, promoting the progression of CVD both in human and animal models [8,51].

## 5. Oxidative-Stress Measurement and d-ROMs Test (Peroxide Assay)

The amount of biologically active ROSs might be an indicator of individual health status, and its quantification holds potential for clinically relevant applications.

Spin trapping and electron spin resonance methodologies are direct and exact methods to quantify ROS [52], but these are not suitable for use in routine clinical practice due to their complexity and high cost. Therefore, the measurement of ROS is mainly based on indirect methods which detect either the final products generated during oxidation processes or intermediate metabolites (reactive oxygen metabolites or ROMs).

Malondialdehyde (MDA) [53], isoprostanes [54], 4-hydroxynonenal (4-HNE) [55], respiratory hydrocarbons [56,57], 8-OH-dG (8-hydroxy-deoxyguanosine) [58] and carbonylated proteins [59] are well-characterised and recognised oxidative-stress biomarkers. They represent end-products of many oxidative reactions, and thus their measurement reflects the oxidative damage that has already occurred.

In contrast, organic peroxides and lipoperoxides (ROMs) [7,60] are intermediate metabolites generated early in the oxidative cascade [7,61]. They still retain a certain degree of reactivity [62], which leads them to further degrade or act as oxidants towards other biomacromolecules [6,60,61]. Hence, the measurement of peroxides and lipoperoxides might be a better indicator of early stages of oxidative-stress damage and might provide an opportunity for a timely intervention (Figure 5).

The d-ROMs test is a colorimetric assay able to detect and quantify circulating organic hydroperoxides (mainly lipoperoxides) [15,63]. It is commonly performed at 37 °C on 10 µL of serum or heparinised plasma obtained from capillary (or venous) sampling. d-ROMs have good stability in long-term-stored blood samples at −80 °C [64]. Since the test principle is based on iron availability, anticoagulants such as EDTA and citrate, or the presence of haemolysis, can interfere with the assay. The test is based on two subsequent reactions: (i) the radicals are formed from the peroxides contained in the sample by oxidising iron in an acid medium (Fenton reaction [24]); (ii) the newly formed radicals oxidise N,N-diethyl-para-phenylendiamine, which turns into a pink colour with a maximum absorbance at 505 nm (Figure 6). The intensity of the colour is proportional to the peroxide concentration. The d-ROMs test has been validated against the gold-standard electron spin resonance (ESR) method, with reference values determined in a study involving over 4000 healthy people [15,16,17,18,19]. The test result is displayed as a “Carratelli Unit” (UCARR), which is an arbitrary unit and 1 UCARR corresponds to the colour development caused by a H_2_O_2_ solution at a concentration of 0.08%. The expected values in a healthy individual are between 250 and 300 UCARR, while higher values denote a surplus of peroxides indicative of a systemic increase in ROS levels. Plasma d-ROMs values can increase in patients repeatedly exposed to high oxygen concentrations [65]. Moreover, d-ROMs have shown a negative correlation with serum creatinine in Japanese men [66], but an effect of renal function on d-ROM levels has not been reported in patients with chronic kidney disease (eGFR < 60 mL/min/1.73 m^2^) [67,68].

## 6. Relation of d-ROMs to Cardiovascular Risk Factors

The relevance of d-ROMS to known CVD risk factors is increasingly being recognised. Although more work is needed on this topic, particularly with respect to clinical outcomes in prospective studies, several meaningful associations have been reported so far.

A Japanese study of 1992 healthy middle-aged subjects found significant associations of serum d-ROMS with age, systolic blood pressure, fasting glucose, and low-density lipoprotein cholesterol (LDL-C); and in a smaller subset of 43 participants, a positive relationship also with high-sensitivity C-reactive protein (hs-CRP) [69]. Furthermore, in 442 subjects who underwent a health check-up investigation, a positive correlation between visceral fat mass and d-ROMs, as well as a negative correlation with measures of kidney function and d-ROMS was reported [66]. Other work that integrated d-ROMS into an oxidative-stress index found that in 160 individuals with diverse ethnicities and diabetes status, d-ROMs were consistently positively associated with BMI, non-high-density lipoprotein cholesterol (non-HDL-C), and fasting glucose [70]. Patients with diabetes, a strong CVD risk factor with rapidly increasing prevalence and incidence in populations worldwide, have substantially higher d-ROM levels compared with controls [71,72]. Specifically in patients with T2DM, another study revealed a positive correlation of circulating d-ROM levels with blood pressure and a negative correlation with HDL-C [71]. Further, in 216 subjects aged over 65 years, higher d-ROMs were significantly correlated with a worse cardio-ankle vascular index (CAVI), a novel physiological parameter proposed to reflect systemic arterial stiffness [73]. These emerging associations clearly support the use of d-ROMs as a clinical biomarker with the capacity to integrate several established cardiovascular risk factors. More results are awaited on a potential prospective association of such correlations with clinical endpoints. Further studies on large prospective cohorts are needed to clarify how d-ROMs and traditional cardiovascular risk factors are related and how d-ROMs can improve cardiovascular risk prediction.

## 7. d-ROM Prognostic Value in Small Cohorts of Individuals with Known Cardiovascular Disease

Extensive preclinical and clinical data support the role of oxidative stress in the complex pathogenesis of atherosclerosis and cardiovascular diseases (CVD) [8,74,75,76]. Moreover, although the direct causality of oxidative stress still remains to be demonstrated, oxidative-stress-related biomarkers have been shown to be independent predictors of cardiovascular events thus providing a potential clinical benefits [8].

Below, we summarise the main results obtained in clinical studies that evaluated the correlations between d-ROM plasma levels and the risk of future CVD events and death in individuals affected by CVD (Table 1).

### 7.1. d-ROMs in Coronary Artery Disease (CAD)

Masaki et al. [77] analysed the correlation between d-ROM values and the risk of future CVD events in a prospective cohort of 265 subjects with stable CVD, of whom 130 had known coronary artery disease CAD (defined as having at least one coronary stenosis proven by coronary angiography (CAG) or history of coronary revascularisation). Study participants were grouped into quartiles of baseline d-ROMs values: (i) 1st-quartile d-ROMs ≤ 285 UCARR (67/34 CAD); (ii) 2nd-quartile 286 UCARR ≤ d-ROMs ≤ 335 UCARR (66/36 CAD); (iii) 3rd-quartile 336 UCARR ≤ d-ROMs ≤ 394 UCARR (67/32 CAD); (iv) 4th-quartile d-ROMs ≥ 395 UCARR (65/28 CAD).

During an average follow-up of 2.66 ± 1.47 years, the subjects in the 4th quartile (d-ROMs ≥ 395 UCARR) showed a higher incidence of all CVD events and death from any cause with respect to the lower quartiles [Hazard Ratio (HR) of the 4th vs. 1st quartile, 10.791 (1.032–112.805), *p* = 0.047]. Furthermore, in the highest quartile of the CAD subgroup, the authors reported a higher incidence of cardiovascular death and all CVD events compared to the 1st and 2nd quartiles (d-ROMs ≤ 335 UCARR), and a significantly higher incidence of major adverse cardiovascular events (MACEs) compared to all other quartiles (vs. 1st, *p* = 0.005; vs. 2nd, *p* = 0.003; vs. 3rd quartile, *p* = 0.019).

The authors concluded that d-ROMs above a cut-off value of 395 UCARR were an independent predictor of future CVD events and death in patients with stable CVD [77].

The potential of d-ROMs as a prognostic biomarker in patients with CAD was also highlighted by Hirata et al. [78]. In their study, d-ROMs were measured in the serum of 395 consecutive patients with angiographically confirmed CAD (coronary artery stenosis ≥75%) and 227 non-CAD subjects (CAD was excluded by CAG or coronary computed tomography). Patients were followed until the occurrence of a CVD event or for an average of 20 months (up to 50 months) [78]. During the follow-up, individuals with d-ROMs >346 UCARR in the CAD group had a higher probability of CVD events than those with d-ROMs ≤ 346 UCARR (*p* = 0.001, log-rank test). At d-ROM levels lower than 346 UCARR, the negative predictive value for having a cardiovascular event was 78.9%. A multivariate Cox hazard analysis identified a d-ROM value > 346 UCARR as a significant and independent predictor of future CVD events (HR = 10.8; 95% CI = 2.76–42.4; *p* = 0.001). Furthermore, when the same authors performed a case-control study comparing d-ROM levels in 163 CAD patients and 163 risk-factor-matched non-CAD controls, d-ROM median values were found to be significantly higher in CAD vs. non-CAD subjects (median d-ROMs = 338 UCARR; IQR = 302–386 UCARR vs. median d-ROMs = 311 UCARR; IQR = 282–352 UCARR; *p* < 0.001). Moreover, in the CAD group, d-ROM levels were also independently correlated with the severity of coronary disease [odds ratio (OR) = 6.15; 95% CI = 1.87–20.3; *p* = 0.003]. Indeed, individuals with single-vessel disease (CAD-SVD) had d-ROMs levels lower than those with multiple-vessel disease (CAD-MVD) (CAD-SVD − median d-ROMs 332 UCARR; IQR = 296–371 UCARR vs. CAD-MVD − median d-ROMs = 360 UCARR; IQR = 313–397 UCARR; *p* < 0.001).

Finally, Hirata et al. [78] measured the d-ROM transcardiac gradient in 90 patients who received CAG and demonstrated that d-ROMs were produced only in the coronary circulation of patients with the presence of CAD, suggesting that coronary atherosclerosis is associated with intracoronary ROS.

Vassalle et al. [80] measured serum d-ROMs in 166 cardiovascular inpatients and followed them for 20 ± 0.3 months. They found that d-ROM values ≥ 482 UCARR (corresponding to the 75th percentile) were a strong and independent predictor of cardiac death (OR = 8.6; 95% CI = 1.5–50.2; *p* = 0.016). The authors then investigated a cohort of 93 subjects with angiographically verified CAD admitted to the coronary care unit and followed them until the occurrence of a cardiovascular event (mean follow-up time 66 ± 28 months) [79]. Kaplan–Meier survival estimates highlighted a worse outcome in patients with elevated d-ROM values (d-ROMs > 481 UCARR). In a multivariate Cox regression analysis, d-ROMs were identified as a significant independent predictor of cardiac death (HR = 3.9; 95% CI = 1.4–11.1; *p* = 0.01), all-cause death (HR = 2.6; 95% CI = 1.1–6.2; *p* = 0.02) and MACEs (HR = 1.8; 95% CI = 1.1–3.1; *p* = 0.03). In line with previous evidence [78], Vassalle et al. [86] confirmed that d-ROM values were higher in CAD vs. non-CAD patients (412 ± 16 UCARR vs. 341 ± 12 UCARR; *p* = 0.004) and that d-ROMs allowed a better stratification of non-CAD, CAD-SVD, and CAD MVD (non-CAD d-ROMs = 341 ± 12 UCARR; CAD-SVD d-ROMs = 377 ± 23 UCARR; CAD-MVD = 434 ± 21 UCARR; *p* = 0.002). Furthermore, when Taguchi et al. [87] examined d-ROM levels in 57 patients with acute coronary syndrome who underwent a repeated intravascular ultrasound (IVUS) study of coronary nonculprit lesions, they reported that d-ROMs represented a discriminative biomarker for coronary plaque progression (HR = 1.018; 95% CI = 1.005–1.032; *p* < 0.01) and plaque destabilisation (HR = 1.022; 95% CI = 1.005–1.038; *p* < 0.01) beyond LDL-cholesterol levels (<100 mg/dl) after 8–10 months of intensive statin treatment. These results suggest d-ROMs as a potential residual risk biomarker in patients with acute coronary syndrome.

Collectively, these data support the potential use of d-ROM values as an oxidative-stress biomarker able to improve risk prediction and stratification in patients with CAD.

### 7.2. d-ROMs in Heart Failure

Hirata et al. [81] measured d-ROMs in 287 patients hospitalised for heart failure with preserved ejection fraction (HFpEF) and 299 subjects without heart failure (non-HF). HFpEF was clinically defined according to the criteria reported by the European Working Group on Myocardial Function [88]. All HFpEF patients were followed up for a mean of 20 months (range 1–50 months) or until the occurrence of a major CVD event. After matching the HFpEF and non-HF groups (*n* = 212 each) for age, gender, and equal incidence of risk factors, d-ROM levels were found to be significantly higher in the HFpEF group than in the non-HF group (median d-ROMs = 343 UCARR; IQR = 312–394 UCARR vs. median d-ROMs = 336 UCARR; IQR = 288–381 UCARR; *p* < 0.001). Furthermore, after HFpEF patients were stratified according to the New York Heart Association (NYHA) [89] functional classes, d-ROM levels were found to be significantly higher in patients with NYHA III/IV than in those with NYHA II (median d-ROMs = 405 UCARR; IQR = 346–478 UCARR vs. median d-ROMs = 338 UCARR; IQR = 308–383 UCARR; *p* < 0.001). During follow-up, the authors divided the HFpEF patients into low- and high-d-ROM groups (median value of d-ROMs 346 UCARR) and observed that the high-d-ROM group had a higher rate of cardiovascular events and hospitalisation for HF (*p* = 0.03). Accordingly, Kaplan–Meier analysis demonstrated a significantly higher probability of CVD events in patients belonging to the high-d-ROM group (log-rank test, *p* = 0.01) and multivariate Cox hazard analysis showed that natural logarithmic-transformed d-ROMs [ln(d-ROM)] were an independent and significant predictor for CVD events (HR = 4.11; 95% CI = 1.04–16.2; *p* = 0.04). At median d-ROMs ≤ 346 UCARR, the negative predictive value for the occurrence of CVD events was 80.1%. A positive correlation between d-ROMs and hs-CRP levels was also identified (r = 0.39; *p* < 0.001), suggesting that oxidative stress and inflammation coexist in HFpEF.

Hitsumoto et al. [82] analysed d-ROMs in elderly patients (mean age = 75 ± 7 years) with chronic heart failure (CHF) without a history of HF hospitalisation, and followed them for a median of 81.1 months (IQR = 6–120 months). The d-ROM median value of 303 UCARR was used as the threshold to divide patients into a low-d-ROM group (group L; mean d-ROMs = 235 ± 45 UCARR) and a high-d-ROM group (group H; mean d-ROMs = 421 ± 85 UCARR). During the follow-up, group H presented a significantly higher incidence of HF hospitalisation than group L (HR = 2.35; 95% CI = 1.37–4.43; *p* < 0.01). Interestingly, the predictive value for the incidence of HF hospital admission was increased by the combination of d-ROMs and brain natriuretic peptide (BNP) levels. In particular, the group with low BNP (<200 pg/mL) but high d-ROMs showed a significantly 2-fold increased risk of HF hospitalisation (HR = 2.21; 95% CI = 1.06–4.71, *p* < 0.05) compared to the group with both low BNP and low d-ROM levels. Furthermore, the group with high BNP (>200 pg/mL) and high d-ROMs exhibited a 4-fold increase in the risk of HF hospital admission compared to patients with either high BNP or high d-ROM levels alone (HR = 9.18; 95% CI = 4.78–22.94; *p* < 0.001). Then, a threshold value of d-ROMs = 319 UCARR allowed the best performance (sensitivity of 86.4% and specificity of 51.8%) in discriminating nonhospitalisation and hospitalisation during the follow-up period.

Finally, the prognostic significance of d-ROMs was recently investigated in patients with heart failure with reduced ejection fraction (HFrEF). Nishihara et al. [84] measured serum d-ROMs in 201 stable patients with left-ventricular ejection fraction (LVEF) ≤50% and followed them for a median of 638 days (IQR = 301–1173 days). The d-ROM median value of 353 UCARR was used to divide HFrEF patients in a high- and low-d-ROMs group (H and L groups, respectively). During the follow-up, the H group (≥353 UCARR) showed a higher rate of total CVD events and hospitalisation for HF decompensation. Consistently, Kaplan–Meier analysis demonstrated a significantly higher probability of HF-related events in HFrEF patients belonging to the H group (log-rank test, *p* = 0.01) and multivariate Cox hazard analysis showed that d-ROMs (per 1 increase) were an independent and significant predictor for CVD events (HR = 1.01; 95% CI = 1.001–1.009; *p* = 0.02). Furthermore, when individuals with HFrEF were stratified according to d-ROM and BNP values, the group with high d-ROMs (≥353 UCARR) and low BNP (≤108 pg/mL) showed a higher occurrence of HF-related events compared to the HFrEF group with low d-ROMs (≤353 UCARR) and low BNP (log-rank test, *p* < 0.01). Noteworthy, the group with high d-ROMs and high BNP (≥108 pg/mL) had the highest event rate. Finally, the author performed receiver operating characteristic (ROC) analysis and confirmed that the d-ROM measurement has an additive value with BNP in the risk stratification of patients with HFrEF (BNP, AUC: 0.694, 95% CI: 0.606–0.782, *p* < 0.001); d-ROMs + BNP, AUC: 0.733, 95% CI: 0.655–0.812, *p* < 0.001).

In summary, d-ROMs might also improve the prediction of future CVD events and HF hospitalisation in patients with chronic heart failure, especially when combined with BNP levels.

### 7.3. d-ROMs in Atrial Fibrillation

Shimano et al. [85] measured d-ROM values in 306 patients with atrial fibrillation (AF) before undergoing radiofrequency (RF) catheter ablation and followed them up for a mean of 1.2 ± 0.8 years, evaluating AF recurrence. In line with previous findings [90], d-ROM levels were found to be significantly higher in patients with persistent AF than in those with paroxysmal AF (341 ± 86 UCARR vs. 305 ± 78 UCARR; *p* < 0.001). Furthermore, a basal d-ROM level > 355 UCARR was a predictor of AF recurrence after pulmonary vein ablation, with a higher rate in the paroxysmal AF group (*p* < 0.01). Thus, d-ROMs may be useful in the management of AF patients for the identification of those at risk for arrhythmia recurrence after catheter ablation.

In conclusion, there is a growing body of evidence supporting the potential clinical use of d-ROMs for cardiovascular risk stratification. The use of d-ROMs as a tool to estimate oxidative stress might be an additive and easy-to-measure prognostic biomarker in patients with known heart disease.

## 8. The d-ROM Prognostic Value in Large General-Population-Based Cohorts

Additional potential prognostic value of d-ROMs has also been assessed in large general-population cohorts. Table 2 summarises the main results of clinical investigations evaluating the correlation of d-ROM levels with the risk of CVD events and mortality in pooled subcohorts of three large European prospective observational studies: the ESTHER [91,92] [*Epidemiologische Studie zu Chancen der Verhütung, Früherkennung und optimierten Therapie chronischer Erkrankungen in der älteren Bevölkerung (German)*], which involved 9940 individuals, male and female, 50–75 years of age, recruited by their general practitioners during a general health check-up; the HAPIEE [93] (*Health, Alcohol and Psychosocial factors In Eastern Europe*), which enrolled 36,500 individuals, men and women aged 45–64, selected in four countries of Central and Eastern Europe, and was designed to investigate the effect of dietary factors, alcohol consumption, and psychosocial factors on health; and the DIANA [94,95] (*Type 2 Diabetes Mellitus: New Approaches to Optimize Medical Care in General Practice*) which involved 1146 German patients with type 2 diabetes (T2DM) with the aim to assess diabetes-related outcomes and possibilities for improvements of health care.

### 8.1. d-ROMs in Individuals with No History of CVD

Xuan et al. [96] performed a nested case-control study based on the 8-year follow-up of the ESTHER [91,92] and the HAPIEE [93] cohorts, which involved Germany, Czech Republic, Poland, and Lithuania. d-ROMs and total thiol levels (TTL), a proxy for reductive capacity [99], were measured in baseline serum samples. Cases were defined as individuals who suffered a nonfatal/fatal myocardial infarction (MI) or nonfatal/fatal stroke during the follow-up. After subjects with history of MI and stroke were excluded, cases and controls were matched (1:5) by cohort, age (± 5 years), and sex, and were combined into a final data set defined as 476 MI cases (with 2380 MI controls) and 454 stroke cases (with 2270 stroke controls).

Baseline d-ROM levels were remarkably higher in MI and stroke cases than in controls. Moreover, individuals with high d-ROM levels (>500 UCARR) had a 2-fold increased risk of MI (OR = 2.04; 95% CI = 1.23–3.37; *p* < 0.05) and a 5-fold increased risk of fatal MI (OR = 5.08; 95% CI = 1.78–14.49; *p* < 0.05). Furthermore, high d-ROMs were also significantly associated with stroke incidence (OR 1.17, 95% CI = 1.01–1.35 for 100 UCARR increase). This correlation between high d-ROMs levels and an increased incidence of MI and stroke was limited to males in a subgroup analysis considering a cut-off value of 400 UCARR.

### 8.2. d-ROMs in the General Population

In a pilot study, Schöttker et al. [97] investigated whether d-ROMs and total thiol levels (TTL) were useful oxidative-stress markers for mortality prediction in 2932 individuals (mean age 70 ± 6) of the ESTHER [91,92] cohort during a mean 3.3 years follow-up. The authors reported that increased d-ROMs were associated with all-cause mortality (HR 1.33; 95% CI = 1.04–1.70 per 100 units increase), but this association was attenuated by inflammation (C-reactive protein levels) and a general morbidity index [97]. However, in a subsequent meta-analysis [68] including 10,622 individuals from the previously mentioned Europe cohort studies, high d-ROM levels were found to be a strong and independent predictor of all-cause mortality (RR 1.20, 95% CI = 1.12–1.29 per 1-Standard deviation (SD) increase; *p* < 0.05), cardiovascular mortality (RR 1.30; 95% CI = 1.12–1.51 per 1-SD increase; *p* < 0.05), and cancer death (RR 1.19; 95% CI = 1.05–1.35 per 1-SD increase; *p* < 0.05) after adjustment for inflammatory status and established risk factors. Furthermore, the predictive value for cardiovascular mortality doubled when d-ROMs were combined with total thiol levels (RR 2.47; 95% CI 1.58–3.98; *p* < 0.05) suggesting the additive benefit of measuring both arms of the Redox balance. Finally, subjects with very high d-ROM levels (d-ROMs > 500 UCARR) not only experienced an elevated risk of short-term mortality, which was 4- to 5-fold higher for cardiovascular and cancer mortality, but the association between d-ROM levels and risk of death remained statistically significant even up to 6 years after baseline.

In summary, high d-ROM levels were found to be a predictor of MI, stroke, premature death, and long-term mortality in the general population.

### 8.3. d-ROMs in Type II Diabetes Mellitus

ROS, including d-ROMs, are increased in people with diabetes mellitus [100,101,102], especially in those with uncontrolled glycemia [103,104,105,106], and oxidative stress might contribute to the increased risk of CVD events and death observed in this population. Particularly, oxidative stress in the vessel wall, caused by a multitude of cardiovascular risk factors, counteracts the beneficial effects of eNOS. Such processes lead to endothelial dysfunction, which is the precursor of arterial stiffness and hypertension, and in itself increases CVD risk [107]. Although beyond the scope of the current review, d-ROMs could be a valuable tool contributing to the identification of patients with endothelial dysfunction [69,108,109,110].

Recently, Xuan et al. [98] have demonstrated the presence of a strong significant prospective association between high d-ROM levels and all-cause mortality in individuals with T2DM. In their study [98], the authors followed 2125 individuals with T2DM from the ESTHER [91,92] and the DIANA [94,95] cohorts over a period of 6–7 years, obtaining oxidative-stress measurements (d-ROMs, TTL, and d-ROMs-to-TTL ratios) at baseline and after 3–4 years.

In the ESTHER [91,92] diabetes subcohort (*n* = 1029), individuals with d-ROM values above 368 UCARR showed an increased all-cause mortality compared to subjects in the lower tertile (HR = 1.67; 95% CI = 1.05–2.67; *p* < 0.05). Furthermore, in the DIANA [94,95] cohort (*n* = 1096), T2DM patients had a prospective association with death when d-ROM values were higher than 450 UCARR (HR = 2.49, 95% CI = 1.74–3.55; *p* < 0.05). Meta-analysis of both cohorts showed that an increase in d-ROM levels corresponded to an augmented risk of all-cause mortality (HR 2.10, 95% CI = 1.43–3.09; *p* < 0.05) and cancer mortality (HR 2.46, 95% CI = 1.44–4.20; *p* < 0.05). However, only high d-ROM-to-TTL ratios were significantly associated with an increased risk of all-cause mortality (d-ROM-to-TTL ratio top tertile; HR 2.50, 95% CI = 1.15–3.65; *p* < 0.05), cardiovascular mortality (d-ROM-to-TTL ratio top tertile; HR 2.57, 95% CI = 1.08–6.14; *p* < 0.05) and incidence of major cardiovascular events (d-ROM-to-TTL ratio top tertile; HR 1.65, 95% CI = 1.07–2.54; *p* < 0.05), pointing out the additive value of combining measurements of oxidative and antioxidant status.

The authors concluded that a strong association existed between oxidative stress and mortality risk in patients with T2DM, and that d-ROMs may be a helpful oxidative-stress biomarker for identifying diabetic patients at higher risk of premature death.

## 9. Discussion

### 9.1. Clinical Perspective

Prevention of future cardiovascular events and death remains one of the major aims of CVD management. Several risk-prediction models have been developed to take into account clinical, genetic, humoral, lifestyle-related, and environmental factors to improve risk stratification both in primary and secondary cardiovascular prevention [111,112,113].

As supported by the available literature, an altered RedOx status is associated with the pathogenesis of CVD from early stages throughout the final diseased state [20]. Therefore, the ability to reliably and easily quantify an imbalance in the RedOx system might become clinically relevant in the risk-stratification process, identifying patients at higher risk, and potentially, those individuals at the earliest stages of disease [8].

Determination of d-ROMs represents a novel oxidative-stress biomarker that, by the indirect quantification of plasma organic peroxides, unveils the systemic increase in ROS levels and the earliest phases of oxidative-stress-induced damage [15,16,17,18,19].

The d-ROMs test has been properly validated using electron spin resonance (ESR), the gold-standard method for assessing free radicals [15,16,17,18,19]. Moreover, d-ROMs have demonstrated a good correlation with other oxidative-stress biomarkers, such as 8-isoprostanes [18]. The advantage of the d-ROMs test is that it can be easily performed in a standard automatic biochemical analyser, but also as a bedside point-of-care test (POCT), thus facilitating its use in clinical practice. Indeed, this POCT uses capillary sampling; thus, no venepuncture is required, and it gives the result in minutes, allowing for timely clinical decision making. However, bedside diagnostic tests present some disadvantages [114]. In fact, POCT is usually performed by non-laboratory-trained individuals, and this may affect the blood testing itself, but also the quality-control procedures for the instrument [115]. Therefore, proper training/recertification of the users, regular monitoring of quality control as described by the manufacturer, and timely quality checks of the instruments are crucial to ensure that the value of POCT results is not diminished. Furthermore, an accurate analysis of costs and benefits should always be considered when performing POCT [114,115,116].

The actual evidence suggests that d-ROMs are related to common CVD risk factors, and that most considerably, high-d-ROM plasma values are an independent predictor of CVD events and mortality, either in individuals with a known or unknown history of CVD (Table 1, Table 2 and Table 3 and Figure 7, Figure 8 and Figure 9).

Risk starts to increase when d-ROM levels are higher than 340 UCARR, and rises considerably above 400 UCARR. Conversely, low d-ROM levels have demonstrated a good negative predictive value for cardiovascular events when measured in small cohorts of patients with CAD and HF [78,81]. Furthermore, the available data indicate that combining d-ROMs with other relevant biomarkers routinely used in clinical practice, such as brain natriuretic peptide (BNP) [82,84], but also with markers of antioxidant capacity (TTL [68,98]), may support a more precise cardiovascular risk assessment.

Finally, several therapeutic interventions (e.g., antihypertensives, beta blockers, statins, and antidiabetic drugs) [117,118,119,120,121,122,123] have been proven to reduce d-ROM levels, suggesting that this oxidative-stress biomarker as a potential parameter for monitoring treatment effects on RedOx status. However, it is still unclear whether reducing d-ROM levels will lead to improved outcomes.

### 9.2. Limitations

Most of the cohorts investigated had a small medium size, and the few larger cohort studies considered had an observational design; thus, confounders and potential information biases cannot be excluded. Secondly, in almost all investigations d-ROMs were determined only at baseline with no repeated measurements during follow-up. However, a significant consensus was found on the ability of d-ROMs to predict future cardiovascular outcomes and mortality, with greater risk corresponding to higher d-ROM baseline levels (Table 3 and Figure 7, Figure 8 and Figure 9). Thirdly, there were differences in the d-ROM risk cut-off values. Indeed, d-ROM cut-offs ranged from 319 to 482 UCARR in patients affected by CVD, while the main d-ROM cut-off value was above 400 UCARR in a general population. Exceptions were the pilot cohort of Schöttker et al. [97] and the ESTHER [91,92] diabetes subcohort of Xuan et al. [98] where the risk of all-cause death increased with average d-ROMs above 380 UCARR and 368 UCARR, respectively (Table 3). However, a limit to the definition of a more robust and narrowed d-ROM cut-off value can be related to the small medium size of the cohorts investigated, and to population biases. In this regard, genetic differences could play a crucial role in the d-ROM cut-off definition [124] as the analysed studies mainly focused on Caucasian and Japanese populations. Moreover, a few investigations [66,79,96] reported differences in d-ROM cut-offs according to sex, with females manifesting higher d-ROM levels than males (female d-ROMs = 365 ± 74 vs. male d-ROMs = 321 ± 61, *p* < 0.0001 [66]; female d-ROMs 463 ± 39 vs. male d-ROMs 392 ± 15, *p* < 0.05 [55]; female d-ROMs 376 ± 50 vs. male d-ROMs 321 ± 39, *p* < 0.001; female d-ROMs 411 ± 44 vs. male d-ROMs 353 ± 41, *p* < 0.001 [96]), thus suggesting that d-ROM cut-off values are also affected by sex-related factors [125,126].

### 9.3. Future Actions

Further longitudinal clinical investigations considering larger cohorts and repeated samples, in combination with meta-analyses, are of paramount importance to enable an international standardisation of the test and a more rigorous identification of adequate cut-off values. Moreover, the cross-comparison of d-ROMs with other oxidative-stress biomarkers and their inclusion in large panels of CVD-related biomarkers will allow a deeper understanding of their clinical significance. Finally, the impact of genetic factors and pharmacological treatments on d-ROM levels should be investigated further to understand the potential usefulness of d-ROMs in driving more personalised therapeutic approaches.

## 10. Conclusions

d-ROM quantification represents a new oxidative-stress-related biomarker with a promising potential for CVD risk stratification in different patient populations, and may provide clinicians with additional information for the management of primary and secondary cardiovascular prevention. In the near future, therapeutic strategies might aim to target oxidative stress in those individuals with high plasma d-ROM levels to positively influence CVD outcomes. Further large prospective clinical investigations are required to support the routine use of d-ROMs in cardiovascular medicine.

## Figures and Tables

**Figure 1 antioxidants-11-01541-f001:**
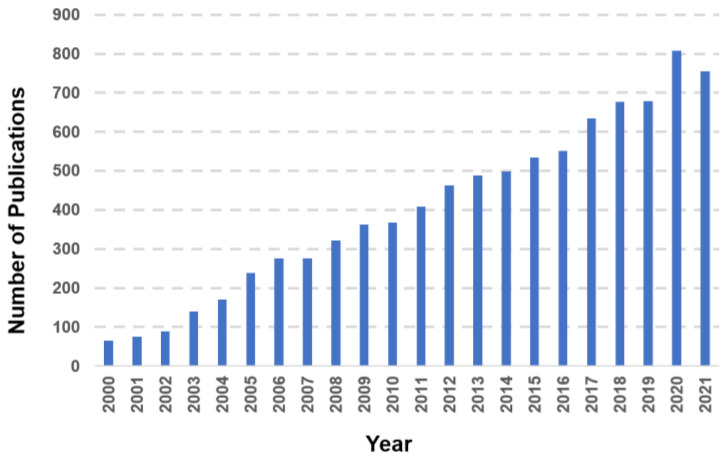
Articles identified on PubMed using “oxidative stress” and “cardiovascular disease” as keywords from 2000 to 2021.

**Figure 2 antioxidants-11-01541-f002:**
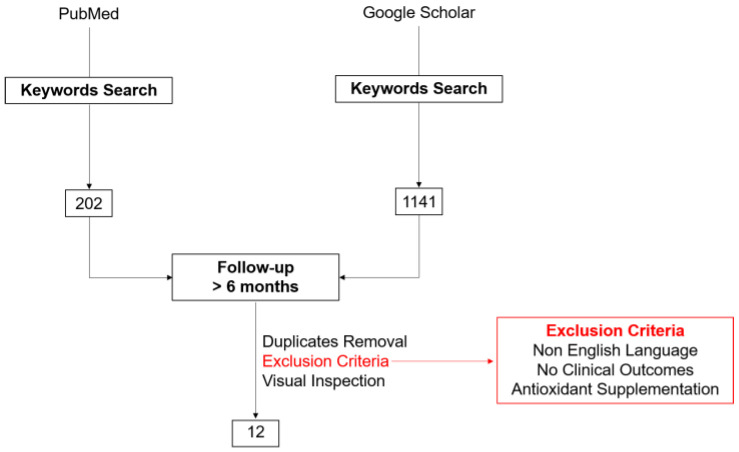
Flow diagram of the literature search.

**Figure 3 antioxidants-11-01541-f003:**
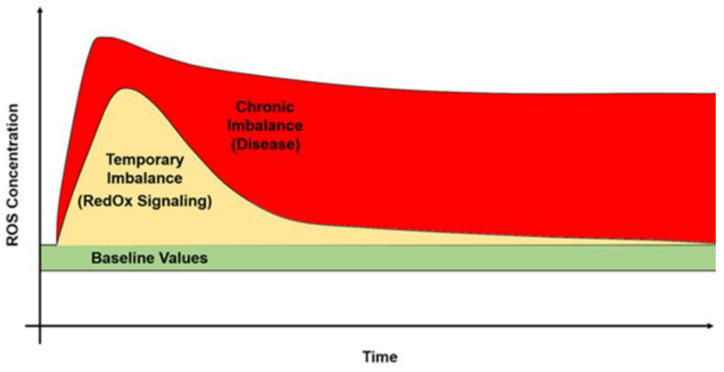
ROS are physiologically produced at low concentration (baseline values, green). ROS have either beneficial effects at middle-low concentration for a short time (orange) or harmful effects at higher concentration for a prolonged time (red).

**Figure 4 antioxidants-11-01541-f004:**
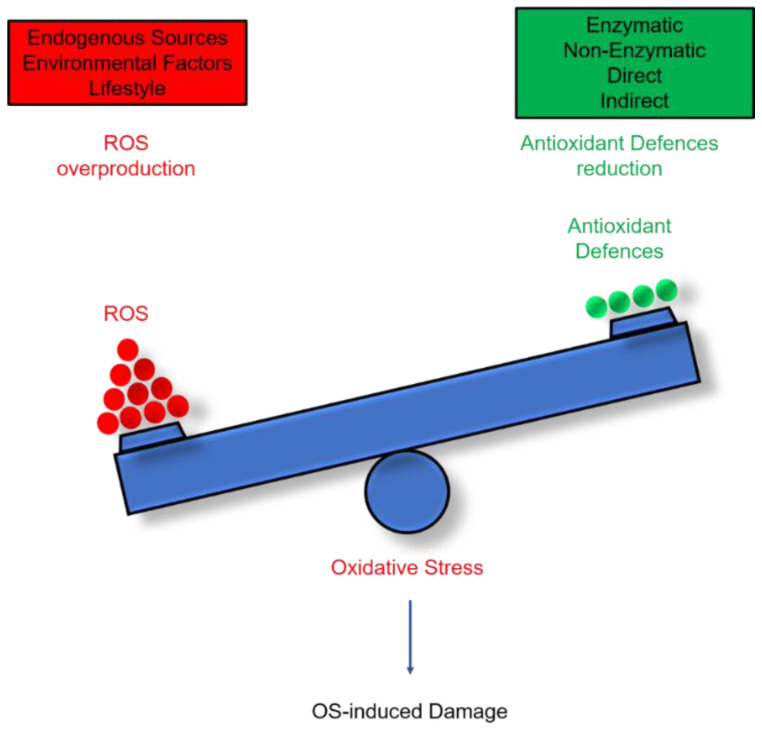
The RedOx balance is preserved by the existing equilibrium between ROS production rate and antioxidant defence systems. Highlighted in red are possible sources of ROS; and in green, the antioxidant defences. Oxidative stress (OS) occurs when there is an overproduction of ROS or a reduced ability of antioxidant defences to counteract the production of ROS. Oxidative stress can induce cellular and tissue injury.

**Figure 5 antioxidants-11-01541-f005:**
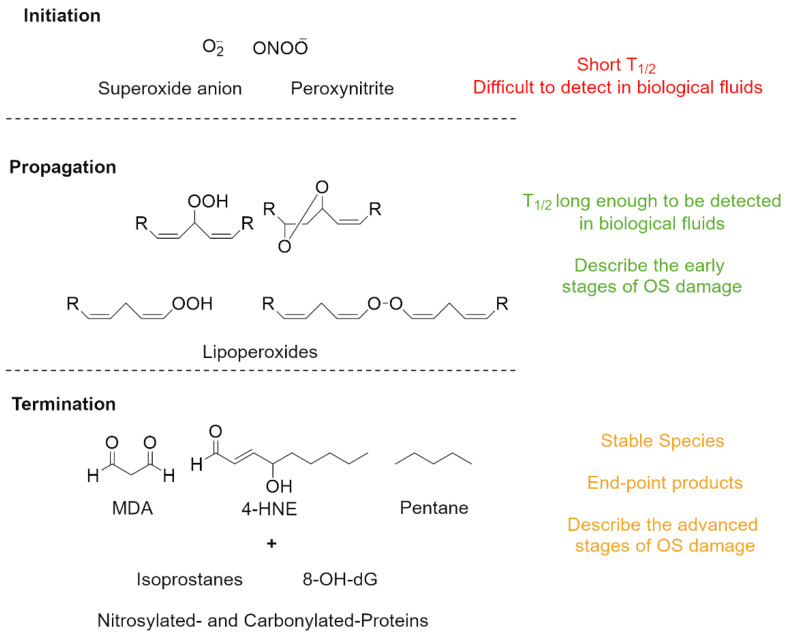
Schematic representation of the oxidative-stress reactive chain with highlighted representative chemical species at the initiation, propagation, and termination stages.

**Figure 6 antioxidants-11-01541-f006:**
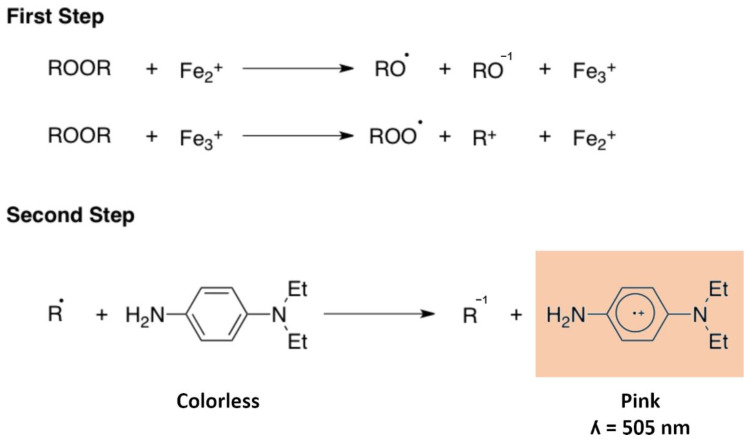
Schematic representation of the sequential reactions occurring in the d-ROMs test.

**Figure 7 antioxidants-11-01541-f007:**
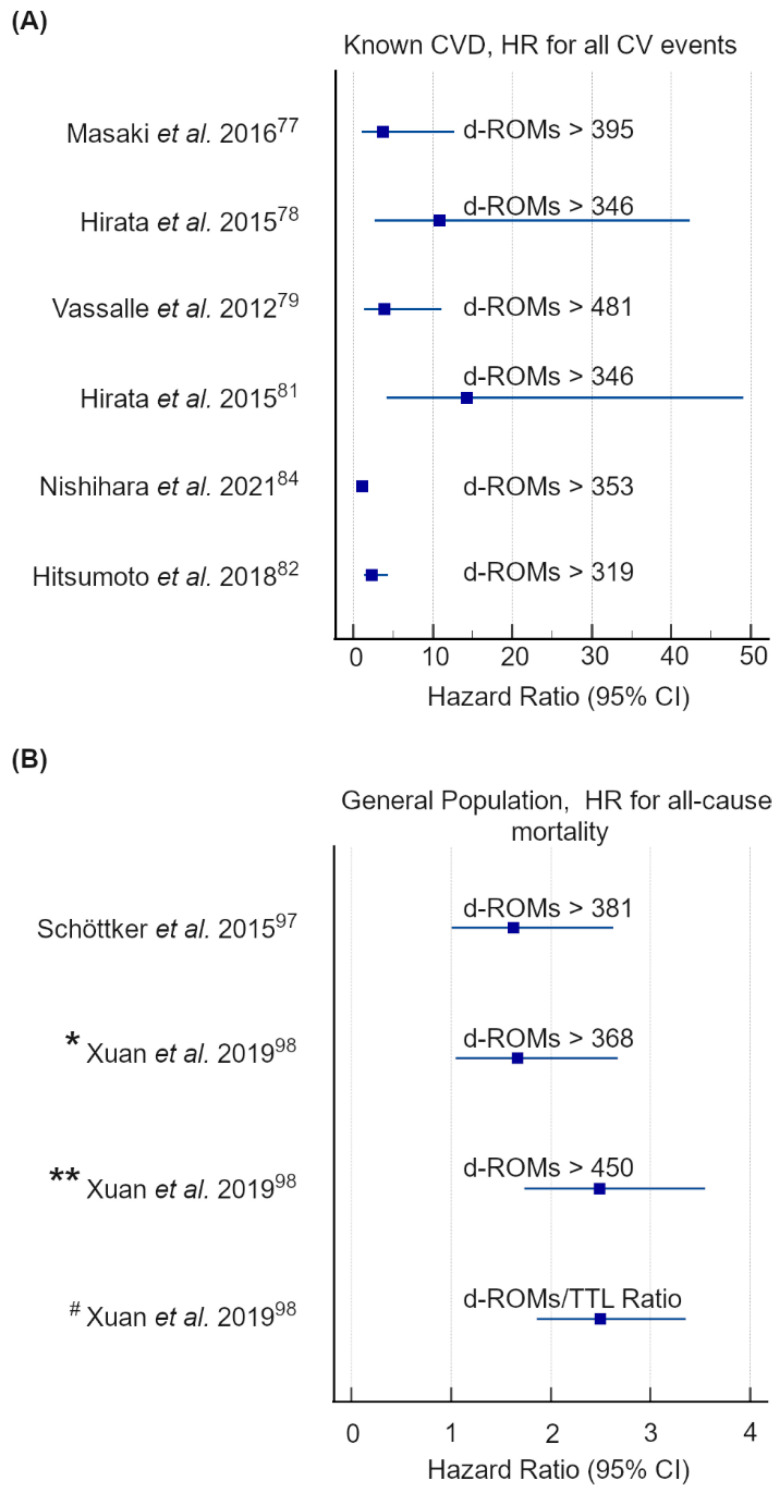
Graphical representation of the data reported in Table 3. The data were grouped depending on population groups (small cohorts of individuals with known cardiovascular disease (CVD) and general-population-based cohorts), clinical outcomes (all cardiovascular (CV) events, cardiovascular mortality, and all-cause mortality) and statistical parameters (HR). (**A**) Hazard ratio and 95% CI for all CV events in small cohorts of individuals with known CVD [77,78,79,81,82,84]; (**B**) hazard ratio and 95% CI for all-cause mortality in general-population-based cohorts [97,98]. * ESTHER cohort; ** DIANA cohort; ^#^ Meta-analysis of ESTHER and DIANA cohorts with HR provided for the d-ROM/TTL ratio top tertile.

**Figure 8 antioxidants-11-01541-f008:**
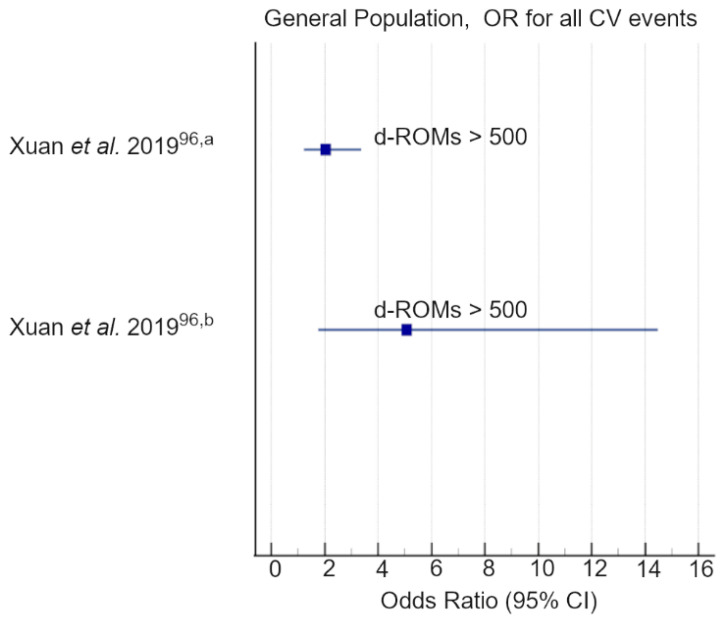
Graphical representation of the data reported in Table 3. The data were grouped depending on clinical outcomes (Xuan et al., 2018 [96] ^a^ OR for myocardial infarction; Xuan et al., 2018 [96] ^b^ OR for fatal myocardial Infarction) and statistical parameters (OR).

**Figure 9 antioxidants-11-01541-f009:**
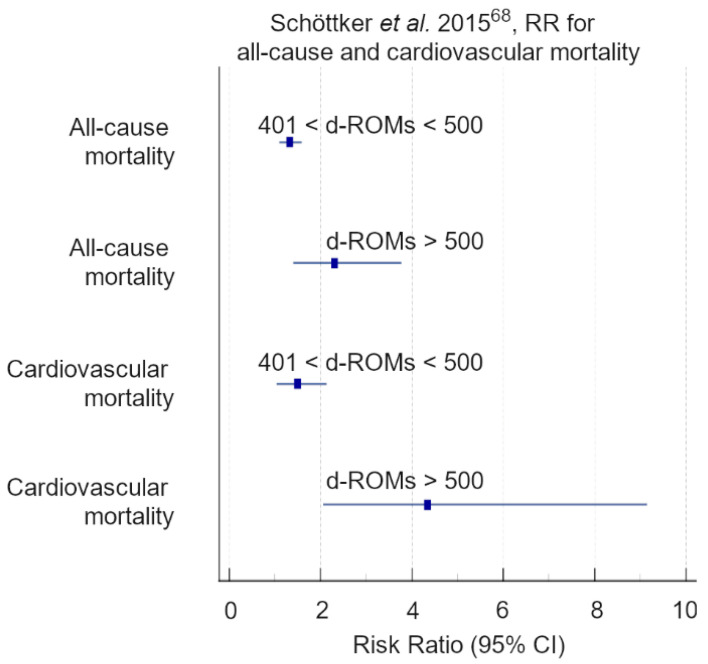
Graphical representation of the data reported in Table 3. The data were grouped depending on clinical outcomes (all-cause mortality and cardiovascular mortality) and statistical parameters (RR), as reported in Schöttker et al., 2015 [68].

**Table 1 antioxidants-11-01541-t001:** List of studies evaluating d-ROM values and the occurrence of CVD events and mortality in individuals with known CVD.

Type of Disease	References	Sample Characteristics			Follow-Up	Main Observations	d-ROM Cut-Off Value (UCARR)
		*n* Age Male/Female		Group Definition Criteria			
**Coronary Artery Disease (CAD)**	Masaki et al., 2016 [77]	All subjects 265 65 ± 13 204/61	CAD group 130 N/D N/D	CAD: Patients with at least one coronary stenosis proven by coronary angiography or past history of coronary revascularisation	2.66 ± 1.47 years	d-ROM values above 395 UCARR were associated with an increased risk of all cardiovascular events * and death for any cause. **In CAD patients, d-ROM values above 395 UCARR were associated with an increase in all cardiovascular events, MACEs ** and cardiovascular death.** * defined as coronary or peripheral revascularisation, cardiovascular surgery, heart failure, and hospitalisation for any cardiovascular cause, in addition to major adverse cardiovascular events (MACEs). ** MACEs defined as death from cardiovascular causes, nonfatal myocardial infarction, and nonfatal cerebral infarction.	**395**
	Hirata et al., 2015 [78]	CAD group 163 69 ± 10 113/50	Non-CAD group 163 69 ± 10 113/50	CAD: Patients with a diameter of stenosis in vessels ≥1.5 mm	Follow-up until the first CVD event or up to 50 months (mean follow-up 20 months)	d-ROM values were significantly higher in risk-factor-matched CAD patients (median = 338 IRQ = 302–386 UCARR) than in risk-factor-matched non-CAD patients (median = 311 IRQ = 282–353 UCARR). Kaplan–Meier analysis showed a **higher probability of CVD events in CAD patients with d-ROMs >346 UCARR.** Multivariate Cox hazard analysis identified ln-DROM as an independent predictor for CVD events.	**346**
	Vassalle et al., 2012 [79]	CAD patients 93 68 ± 10 75/18		CAD: Patients with angiographically documented CAD	66 ± 28 months	Kaplan–Meier survival estimates showed a **significantly worse outcome in patients presenting elevated d-ROM level (>75th percentile, corresponding to 481 UCARR) for cardiac death, all-cause death, and MACEs.** In a multivariate Cox regression model, an elevated oxidative stress remained a significant predictor of cardiac and all-cause death.	**481**
**CVD**	Vassalle et al., 2006 [80]	166 cardiovascular inpatients 66 ± 1 112/45		Investigated in a clinical cardiology setting	Follow-up for 20 ± 0.3 months	**d-ROM values ≥ 482 UCARR** (corresponding to the 75th percentile) **were a strong and independent predictor of cardiac death and total mortality.**	**482**
**Heart Failure (HF)**	Hirata et al., 2015 [81]	HFpEF group 212 70 ± 9 127/85	Control Group 212 70 ± 9 136/76	HFpEF: patients with symptoms of HF or mildly reduced left-ventricular systolic function (LVEF > 50% and left-ventricular end-diastolic volume index <97 mL/m^2^ and evidence of abnormal left-ventricular diastolic distensibility and stiffness)	Patients followed up to the first CVD events or up to 50 months (mean follow-up 20 months)	d-ROM levels were significantly higher in risk-factor-matched HFpEF patients (median = 343 IRQ = 312–394 UCARR) than in non-HF controls (median = 336 IRQ = 288–381 UCARR). Authors used **median value of d-ROMs (346 UCARR) to divide HFpEF patients into low- and high-d-ROM groups.** Total CVD events and hospitalisation for HF were significantly higher in high-d-ROM group than in low-d-ROM group. Kaplan–Meier analysis demonstrated a **significantly higher probability of CVD events in HFpEF patients with high-d-ROMs than in those with low-d-ROMs.**	**346**
	Hitsumoto et al., 2018 [82]	Patients with chronic heart failure (CHF)			81 months (range, 6–120 months)	The mean value for Low d-ROM group was 235 ± 45 UCARR and for **High d-ROM group was 421 ± 81 UCARR** Multivariate Cox regression analysis revealed that the **High d-ROM group exhibited a significantly higher risk for HF hospitalisation than Low d-ROM group.** 319 UCARR was considered as best value for discriminating between non-hospitalisation and hospitalisation for HF during follow up.	**319**
		L group (d-ROMs < 303 UCARR) 214 74 ± 6 58/156	H group (d-ROMs > 303 UCARR) 214 76 ± 8 50/164	CHF was defined according to the ACC/AHA 2005 Guidelines for the Diagnosis and Management of Heart Failure in Adults [83]			
	Nishihara et al., 2021 [84]	Patients with HFrEF (201) and without HF (241)		HFrEF: patients with Framingham criteria for congestive HF with left-ventricular ejection fraction <50%, in stable conditions after optimal medical therapy	mean follow-up 638 days (IQR, 301–1173 days)	d-ROM levels were significantly higher in HFrEF patients (median = 344 IRQ = 297–390 UCARR) than in risk-factor-matched non-HF controls (median = 323 IRQ = 282–366 UCARR) Authors used **median value of d-ROMs (353 UCARR)** to divide HFrEF patients into low- and high-d-ROM groups. **Total CVD events and hospitalisation for HF were significantly higher in high-d-ROM group** than in low-d-ROMs group. Kaplan–Meier analysis demonstrated a significantly higher probability of HF-related events in HFrEF patients with high-d-ROMs than in those with low-d-ROMs.	**353**
		L group (d-ROMs < 353 UCARR) 100 69.2 ± 9.7 100/201	H group (d-ROMs > 353 UCARR) 101 69.2 ± 9.7 101/201				
**Atrial Fibrillation (AF)**	Shimano et al., 2009 [85]	Paroxysmal AF group 225 59 ± 11 162/63	Persistent AF group 81 59 ± 12 64/17	Patients with paroxysmal AF or persistent AF admitted for elective radiofrequency (RF) catheter ablation. Patients undergoing haemodialysis and those with structural heart disease were excluded.	1.2 ± 0.8 years	d-ROM levels in patients with persistent AF (341 ± 85 UCARR) were significantly higher than in patients with paroxysmal AF (305 ± 78 UCARR). Kaplan–Meier analysis revealed that **the highest quartile of basal d-ROM levels (>355 UCARR),** but not within the highest quartile of hs-CRP levels (>1.20 mg/L), **exhibited a significantly higher AF recurrence rate after radiofrequency catheter ablation, in the paroxysmal AF group.**	**355**

**Table 2 antioxidants-11-01541-t002:** List of studies analysing the correlation of d-ROM values with CVD events and mortality in large population-based cohorts.

References	Sample	Follow Up	Main Observations
	*n* Age Male/Female		
XUAN et al., 2019 [96]	MI group 476 cases (age median = 64 IRQ = 58–69, 334 males and 142 females); 2380 controls (age median = 64 IRQ = 58–69, 1680 males and 700 females) Stroke group 454 cases (age median = 66 IRQ = 61–70, 239 males and 215 females); 2270 controls (age median = 66 IRQ = 61–69, 1195 males and 1075 females)	8 years	d-ROM levels were statistically significantly higher among MI cases than controls, and d-ROM levels were statistically significantly associated with total MI incidence. **A strong 5-fold increased risk of fatal MI was observed for d-ROM values > 500 UCARR.** **d-ROM levels were associated with both MI** (OR = 1.21; 95% CI = 1.05–1.40; for 100 UCARR increase) **and stroke incidence** (OR = 1.17; 95% CI = 1.01–1.35; for 100 UCARR increase). The observed relationships were stronger with fatal than with nonfatal endpoints.
SCHÖTTKER et al., 2015 [97]	2932 70 ± 6 1321/1611	3.3 ± 0.7 years	**The top tertile of d-ROM levels (>381 UCARR),** compared with the bottom tertile, **was associated with all-cause mortality,** in models adjusted for age, sex, education, smoking, physical activity, and alcohol consumption.
SCHÖTTKER et al., 2015 [68]	1702 cases of death and 8310 controls divided in 4 cohorts age range: 45–85 years male% range cases: 59–68 male% range controls: 43–68	from 6 to 8 years	d-ROMs were significantly associated with all-cause mortality independently from established risk factors (including inflammation). Regarding cause-specific mortality, compared to low d-ROM levels (≤340 UCARR), **very high d-ROM levels (>500 UCARR) were strongly associated with both cardiovascular** (relative risk (RR), 5.09; 95% CI, 2.67–9.69) **and cancer mortality** (RR, 4.34; 95% CI, 2.31–8.16).
XUAN et al., 2019 [98]	2125 patients with T2DM from ESTHER and DIANA cohorts. In total, 205, 179, and 394 MCE ^#^, cancer, and all-cause mortality cases were observed. ESTHER: 1029 (535males, 494 females) DIANA: 1096 (600 males, 496 females)	from 6 to 7 years	**An increase in d-ROM levels corresponded to an increased risk of all-cause mortality, in particular in T2DM males with previous medical history of CAD.** However, only **high d-ROM-to-TTL ratios were statistically significantly associated with both an increased all-cause mortality** (d-ROMs-to-TTL ratio top tertile; HR 2.50, 95% CI = 1.86–3.36; *p* < 0.05) **and incidence of major cardiovascular events** (d-ROMs-to-TTL ratio top tertile; HR 1.65, 95% CI = 1.07–2.54; *p* < 0.05).

^#^ MCE (major cardiovascular events: myocardial infarction, stroke, and cardiovascular mortality).

**Table 3 antioxidants-11-01541-t003:** Summary of the studies analysing the association between d-ROM values and the risk of CVD events and mortality.

Chapter	References	Hazard Ratio (HR), Odds Ratio (OR) or Risk Ratio (RR) (Confidence Interval 95%)	Event	Population Size	d-ROM Cut-Off (UCARR)
**d-ROM prognostic value in small cohorts of individuals with known CVD**	Masaki et al., 2016 [77]	(HR) 3.755 (1.108–12.730), *p* = 0.034	CVD events	265	395
	Hirata et al., 2015 [78]	(HR) 10.8 (2.76–42.4), *p* = 0.001	CVD events	395	346
	Vassalle et al., 2006 [80]	(OR) 8.6 (1.5–50.2), *p* = 0.016	Cardiac death	166	482
	Vassalle et al., 2012 [79]	(HR) 3.9 (1.4–11.1), *p* < 0.01	Cardiac death, MACEs, all-cause death	93	481
	Hirata et al., 2015 [81]	(HR) 14.3 (4.19–49.1), *p* < 0.001	CVD events	287	346
	Nishihara et al., 2021 [84]	(HR) 1.01 (1.001–1.009), *p* = 0.02	CVD events and HF-related events	201	353
	Hitsumoto et al., 2018 [82]	(HR) 2.35 (1.37–4.43), *p* < 0.01	Heart failure hospitalisation	428	319
**d-ROM prognostic value in general population-based cohorts**	Xuan et al., 2019 [96]	(OR) 2.04 (1.23; 3.37), *p* < 0.05	Myocardial infarction (MI)	2856	500
		(OR) 5.08 (1.78; 14.49), *p* < 0.05	fatal MI		500
		(OR) 1.21 (1.05–1.40), *p* < 0.05	MI odds ratio for 100 UCARR increase		-
		(OR) 1.17 (1.01–1.35), *p* < 0.05	Stroke odds ratio for 100 UCARR increase		-
	Schöttker et al., 2015 [97]	(HR) 1.63 (1.01; 2.63), *p* < 0.05	All-cause death	2932	381
		(HR) 1.33 (1.04; 1.70), *p* < 0.05	All-cause death per 100 UCARR increase		-
	Schöttker et al., 2015 [68]	(RR) 1.32 (1.10–1.59), *p* < 0.05	All-cause mortality	10,012	401–500
		(RR) 2.30 (1.40–3.77), *p* < 0.05			>500
		(RR) 1.49 (1.04–2.13), *p* < 0.05	Cardiovascular mortality		401–500
		(RR) 4.34 (2.06–9.15) *p* < 0.05			>500
	Xuan et al., 2019 [98]	(HR) 1.67 (1.05–2.67), *p* < 0.05	All-cause mortality	1029 *	368
		(HR) 2.49 (1.74–3.55) *p* < 0.05		1096 **	450
		(HR ^§^) 2.50 (1.86–3.36), *p* < 0.05	All-cause mortality	2125 ***	-
		(HR ^§^) 1.65 (1.07–2.54), *p* < 0.05	MCE ^#^		-

* ESTHER cohort; ** DIANA cohort; *** meta-analysis, ^§^ HR is provided for the d-ROM/TTL ratio top tertile; ^#^ MCE (myocardial infarction, stroke, and cardiovascular mortality).

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
