# Peer review of "The Prognostic Value of Derivatives-Reactive Oxygen Metabolites (d-ROMs) for Cardiovascular Disease Events and Mortality: A Review"

_antioxidants, 2022, doi:10.3390/antiox11081541_

Round 1

Reviewer 1 Report

In the paper "The prognostic value of derivatives-Reactive Oxygen Metabo lites (d-ROMs) for cardiovascular disease events and mortality. 3 A review", the authors performed a review of the prognostic value prognostic value of d-ROMs for cardiovascular disease events and mortality in individuals with known and unknown cardiovascular diseases. 

The research methodology of the review is correct, the paper is well written and organized,
the figures are excellent and overall the paper  is of course of interest for the reader of Antioxidants. 

I congratulate the authors for the excellent paper and recommended the acceptance of the review.   

Author Response

Dear Reviewer 1,

Many thanks for the time you have spent in reading and reviewing our manuscript. Your comments and your positive assessment of our work are much appreciated.

Reviewer 2 Report

Very interesting and clear review on the role of d-ROMs as biomarkers in cardiovascular disease.

Please clarify the methodology for the review: were  the PRISMA guidelines observed and applied? Please inclusive the information in the manuscript.

Author Response

Dear Reviewer 2,

Many thanks for the time you have dedicate to reading and reviewing our manuscript. Your positive feedback is much appreciated.

Point 1: Please clarify the methodology for the review: were the PRISMA guidelines observed and applied? Please inclusive the information in the manuscript.

Response 1: Our review is a narrative review, thus the full PRISMA (Preferred Reporting Items for Systematic Reviews and Meta-Analyses) guidelines have not been applied. However, a flow diagram has been added to the Literature Search Methods (section 2 of the review).

Reviewer 3 Report

In the present paper, Pigazzani et al. conducted a comprehensive study evaluating the usefulness of circulating derivatives-Reactive Oxygen Metabolites (d-ROMs) levels as a potential clinical marker for the prognosis of cardiovascular events and mortality in patients with asymptomatic and symptomatic cardiovascular disease (CVD). Since d-ROMs are an emerging biomarker of oxidative stress easily measurable, this well-written review is of particular interest. I only have the following specific comments:

-       The manuscript systematically analyses most of the research evidence in the literature rather than summarises the role of d-ROMs as prognostic biomarkers of CVD. My suggestion would be to convert and format it as a systematic literature review.

-        A flow diagram showing the study selection should be added. It is also recommended that the authors revise whether any piece of evidence is lacking or not. 

-        Graphically displaying the data of each study in the form of a forest plot merits attention.

-        The use of bedside point-of-care in vitro diagnostic tests has some challenges. This issue should be explained and discussed.

-        Potential publication bias as well as heterogeneity should be assessed and reported.

-        Endothelial dysfunction, as a key driver of oxidative stress (PMID: 32282914), should be further addressed.

-        Similarly, oxidative stress is a hallmark of major cardiovascular risk factors. I would suggest expand the analysis to the effect of cardiovascular risk factors on this oxidative stress marker.

-        In my opinion, authors should emphasize the clinical impact of the study.

Author Response

In the present paper, Pigazzani et al. conducted a comprehensive study evaluating the usefulness of circulating derivatives-Reactive Oxygen Metabolites (d-ROMs) levels as a potential clinical marker for the prognosis of cardiovascular events and mortality in patients with asymptomatic and symptomatic cardiovascular disease (CVD). Since d-ROMs are an emerging biomarker of oxidative stress easily measurable, this well-written review is of particular interest.

Dear Reviewer 3,

Many thanks for the time you have spent in reading and reviewing our manuscript. Your positive comments are much appreciated. Please, find below our responses:

I only have the following specific comments:

Point 1: The manuscript systematically analyses most of the research evidence in the literature rather than summarises the role of d-ROMs as prognostic biomarkers of CVD. My suggestion would be to convert and format it as a systematic literature review.

Response 1: Thank you for your suggestion to reformat our review as a systematic review. We know that this would help reduce the risk of bias and make findings more reliable. However, this review has been thought and organized as a narrative review because the evidence available on the prognostic value of d-ROMs is still sparse and based on small/medium size cohorts. We feel that a systematic review with a metanalysis of the data will be more appropriate at a later time point when longitudinal clinical studies with outcomes on d-ROMS (with repeated samples) will become available. We hope that you can consent to our reasoning.

Point 2: A flow diagram showing the study selection should be added. It is also recommended that the authors revise whether any piece of evidence is lacking or not.

Response 2: Thank you for pointing this out. We have added a flow diagram showing the study selection in the section 2 of the manuscript.

Point 3: Graphically displaying the data of each study in the form of a forest plot merits attention.

Response 3: Thank you for your suggestions. Forest plots are useful charts in systematic review and/or meta-analysis studies. Our study is a narrative review, thus, as per your suggestion we have  provided a visual overview of the data reported in Table 3. We believe that this will help the reader in having an easier comprehension of the data. We have added to the manuscript Figure 7, Figure 8 and Figure 9. The data were grouped depending on population groups (small cohorts of individuals with known CVD and general population-based cohorts), clinical outcomes (all cardiovascular events and all-cause mortality) and statistical parameters (HR, OR and RR).

Point 4: The use of bedside point-of-care in vitro diagnostic tests has some challenges. This issue should be explained and discussed.

Response 4: We agree with this point, indeed Point of Care Testing (POCT) has pros but also cons.

The main benefit is that the results are available while the patient is still in the clinic, thus allowing clinical decisions to be made immediately. Also POCT requires less blood volume and many use capillary testing, so no venipuncture is required. Therefore, the patient does not need to attend a phlebotomist for blood draw (this is also less invasive for the patient). Yet, POCT testing can provide vital information in minutes allowing key choices in patient care to be made.

However, the main caveats with POCT are that the test may be performed by busy members of the clinical team and not by trained laboratory technicians. This may affect not only the blood testing itself but also the quality control procedures for the instrument. POCT testing is also usually more expensive than sending blood for testing to the routine hospital laboratory.

To prevent issues with testing and quality of the test and the instrument, it is crucial that all users have been trained, run quality controls as described by the manufacturer and the POCT instrument itself undergoes servicing and manufacturing quality checks regularly. Pre-analytical processes must also be taken into consideration when individuals are being trained.

We have added the following sentence to discussion paragraph regarding POCT: “However, there are challenges with POCT in terms of personnel training, quality control and costs”. 

Point 5: Potential publication bias as well as heterogeneity should be assessed and reported.

Response 5: Thank you for this suggestion, however, as per our response in point 1, we have not applied these aspects to our narrative review as these are tools only used in systematic reviews.

Point 6: Endothelial dysfunction, as a key driver of oxidative stress (PMID: 32282914), should be further addressed.

Response 6: You have raised an important point here. Indeed, the vascular endothelium is a major target of oxidative stress and d-ROMs has been reported as a useful marker for endothelial damage (PMID 21822395). Moreover, a negative correlation has been shown between d-ROMs and Flo-mediated vasodilation (PMID 26105396).

This topic is really important and it would have been interesting to address it, however, it would have been out of scope for this particular review which primarily aimed to collect and analyse the available evidence on the ability of d-ROMs to predict clinical outcomes.

Point 7:  Similarly, oxidative stress is a hallmark of major cardiovascular risk factors. I would suggest expand the analysis to the effect of cardiovascular risk factors on this oxidative stress marker.

Response 7: We agree that oxidative stress is a hallmark of major cardiovascular risk factors. Indeed, the available literature suggests that a correlation exists between d-ROMs levels and traditional cardiovascular risk factors. However, this topic would have been out of scope for this review as explained in Response 6. A new manuscript has been planned to specifically address this relevant topic.

Point 8: In my opinion, authors should emphasize the clinical impact of the study.

Response 8: Thank you for this suggestion. We haven’t emphasized too much the clinical impact of our study because, the available evidence is good but still not strong enough to change the daily clinical practice. However, the test is available and it is validated, thus clinicians can start using it to generate the evidence we need to confirm our results. Moreover, researchers could start using it in clinical trials to verify the consistency and usefulness of the test.

We think that this study is important because it shows that there exists an easy way of assessing oxidative stress in the clinic and this could potentially improve the risk stratification of patients with cardiovascular diseases (in particular CAD and heart failure) but even of subjects without cardiovascular disease. Moreover, it would be interesting to verify, if a reduction in oxidative stress can lead to a reduction of cardiovascular outcomes. If this was the case, we should start using treatments able to reduce oxidative stress (e.g. statins) and d-ROMs could be used to follow-up patients.

As per the reviewer’s suggestion we have amended the Conclusion section to emphasize the potential clinical impact of d-ROMs as follows:

d-ROMs quantification represents a new oxidative stress-related biomarker with a promising potential for CVD risk stratification in different patient populations and may provide clinicians with an additional information for management of primary and secondary cardiovascular prevention. In the near future, therapeutic strategies might aim to target oxidative stress in those individuals with high plasma d-ROMs levels to positively influence CVD outcomes. Further large prospective clinical investigations are required to support the routine use of d-ROMs in cardiovascular medicine.